# On the Electrical and Optical Properties Stability of P3HT Thin Films Sensitized with Nitromethane Ferric Chloride Solutions

**Laura Hrostea** [1,2,*] , **Liviu Leontie** [1], **Marius Dobromir** [3], **Corneliu Doroftei** [4] **and Mihaela Girtan** [2,*]

[1]  Faculty of Physics, Alexandru Ioan Cuza University of Iasi, Bulevardul Carol I, nr. 11, 700506 Iasi, Romania; lleontie@uaic.ro

[2]  Photonics Laboratory, (LPhiA) E.A. 4464, SFR Matrix, Faculty of Sciences, Angers University, 2 Bd Lavoisier, 49000 Angers, France

[3]  Interdisciplinary Research Institute, Sciences Department, Alexandru Ioan Cuza University of Iasi, 11 Carol I Blvd., 700506 Iasi, Romania; marius.dobromir@uaic.ro

[4]  Institute of Interdisciplinary Research, Alexandru Ioan Cuza University of Iasi, Integrated Center for Studies in Environmental Science for North-East Region, Bulevardul Carol I, nr. 11, 700506 Iasi, Romania; corneliu.doroftei@uaic.ro

*  Correspondence: laura.hrostea@etud.univ-angers.fr (L.H.); mihaela.girtan@univ-angers.fr (M.G.)

**Abstract:** The electrical and optical properties stability of poly(3-hexylthiophene) (P3HT) thin films sensitized with nitromethane ferric chloride ($FeCl_3$) solution was investigated. The optical properties modifications were studied by spectrophotometry and ellipsometry. For electrical characterizations, electrical resistivity measurements were performed. In agreement with the observations of other authors, an important decrease in the electrical resistivity by six orders of magnitude was noticed. In addition, the repeatability and stability of this phenomenon were investigated over a few weeks after sensitization and during different cycles of heating and cooling, both in the dark and under illumination.

**Keywords:** polymer; sensitization; electrical resistivity; solar cell

## 1. Introduction

Organic photovoltaics (OPV) has attracted a particular research interest over the last twenty years, due to important manufacturing advantages of polymer materials, such as high molecular diversity, mechanical flexibility, low-cost of manufacturing, feasibility, easy processability, etc. [1–7]. Polymer–fullerene solar cells (with a polymer as donor material and fullerene as an acceptor material) are one of the most widely studied solar cells up to today [8,9], achieving an efficiency of almost 10% [10]. More recently, new types of acceptor materials were studied in order to replace fullerenes in the active material blend. These new organic solar cells were called "non-fullerene solar cells", and the highest efficiency obtained, of over 15%, was reported in 2018, by researchers from South China University of Technology (SCUT) [11,12]. As donor polymers, the new very promising candidates in the last four years are Poly[(2,6-(4,8-bis(5-(2-ethylhexyl)thiophen-2-yl)-benzo [1,2-b:4,5-b']dithiophene))-alt-(5,5-(1',3'-di-2-thienyl-5',7'-bis(2-ethylhexyl)benzo[1',2'-c:4',5'-c'] dithiophene-4,8-dione)] (PBDB-T) or its derivatives, Poly(9,9-dioctylfluorene-*alt*-bithiophene) (F8T2) or Poly[N-9'-heptadecanyl-2,7-carbazole-alt-5,5-(4',7'-di-2-thienyl-2',1',3'-benzothiadiazole)] (PCDTBT). Nevertheless, the poly(3-hexylthiophene) (P3HT) is still an intensively used polymer.

In order to improve the electrical conductivity, sensitization with nitromethane ferric chloride (FeCl$_3$) solutions was proposed in References [13–15]. Being a cheap chemical oxidation agent, FeCl$_3$ is able to diffuse through polymer chains, acting like a p-type material. The physical properties of polymer can be modified because of the π-electron delocalization, allowing intercalation of new molecules between the polymer chains, without damaging the initial structure [13]. The efficiency of this sensitization process is related to the polymer thin-film thickness [16–19]. Depending on FeCl$_3$ solution concentration in nitromethane, a visible decrease in electrical resistivity can be obtained, compared to the value of pristine polymer [20]. However, above a solution saturation threshold, the additional amount is converted in traps or obstacles for charge carriers transport [21].

Despite these advantages, one of the general main problem of the development of OPV is the poor stability of films and devices. In this regard, the aim of this paper is to investigate the reproducibility and the stability over time and during cooling and heating cycles, in dark and under illumination, of the electrical properties of sensitized polymer films.

## 2. Materials and Methods

Regioregular (>94.7%) P3HT with an average molecular weight of 34,100 was purchased from Ossila. The polymer was dissolved in chlorobenzene (16 g/L). The investigated films were deposited by spin coating, at a spinning speed between 600 and 800 rpm, on glass and patterned indium tin oxide (ITO) substrates. The P3HT polymer thin films measured 90 ± 5 nm in thickness. ITO-coated glass substrates with a sheet resistance of 15 Ω/□ were purchased from Kintec. After deposition, samples were dried in an oven, for 1 h, at 100 °C. The sensitization with FeCl$_3$ was done as described in Reference [22], by samples submersion in a 5 g/L nitromethane solution, for one hour. The submersed sample was dried freely in the normal atmosphere, and no chemical agent surplus cleaning procedures were applied. Samples were kept in normal atmosphere, at room temperature, and under dark conditions.

The thickness was measured by profilometry, using a Veeco Dektak 6M Stylus instrument (Veeco, Munich, Germany). The optical properties were investigated by using a two-beam spectrophotometer UV/VIS S9000 (Labomoderne, Gennevilliers, France) in a wavelength range of 250–1100 nm and by spectroscopic ellipsometry, at room temperature, and in the same spectral range, using an UVISEL ellipsometer (Horiba Jobin Yvon, Longjumeau, France), involving a 75 W high discharge Xenon lamp as light source, at an incidence angle of 70°. The optical constants, refractive index, and extinction coefficient, were determined. The structural properties were investigated, using a Shimadzu LabX XRD 6000 diffractometer (CuKα radiation, wavelength λ = 1.54182 Å, Shimadzu, Columbia, MD, USA). The electrical measurements were performed in the configuration given in Figure 1, using a regulated DC power supply and an Agilent 4339B high-resistance meter. The temperature was varied between 30 and 100 °C, during several consecutive heating/cooling cycles, under dark conditions and under white-light illumination. Higher temperature values were avoided, in order not to damage the polymer.

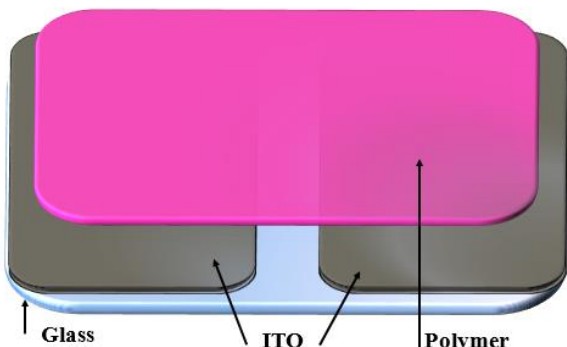

**Figure 1.** Sample configuration (used for electrical measurements). Reprinted with permission from [20]; Copyright 2013 Elsevier.

## 3. Results

Crystal structure analysis by X-ray diffraction was performed for pristine and sensitized P3HT thin films. A small peak (Figure 2), located at 2θ = 5.28°, was emphasized, indicating a small local order of the polymer and suggesting a possible (100) edge-on lattice plane of thiophene orientation [23].

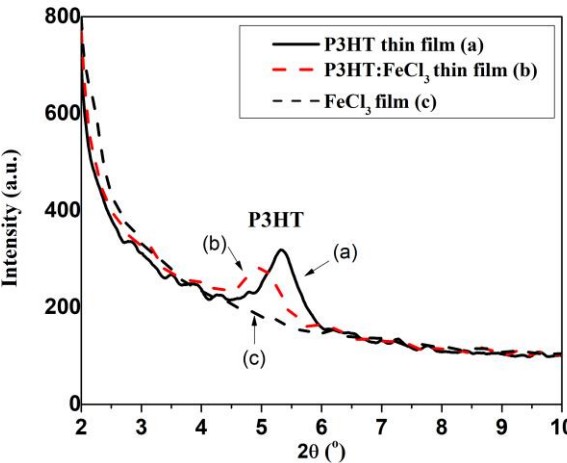

**Figure 2.** XRD patterns of sensitizing agent and polymer thin films.

The reduction of the peak intensity suggests a reduction of the local order after sensitization, and the small shift is in agreement with the observation of Reference [23].

In Figure 3, typical absorption spectra of pristine polymer films (continuous black line), sensitized polymer films (dashed red line), and FeCl$_3$ films (dotted black line), used as reference, are presented. Caused by π–π* transitions in conjugated π systems [24], the absorption peaks specific to P3HT polymer are emphasized at 528 and 554 nm, continued by a shoulder at 600 nm. Compared with the absorption spectrum of the ferric chloride, which exhibits a weak absorption peak in the wavelength range up to 400 nm, the ferric chloride molecules are inserted in the polymer film, interacting with the polymeric chains. In the case of P3HT:FeCl$_3$ thin film, the spectrum is changed, displaying a rapid decrease in the polymer absorption in the visible range, but an enhancement in near-infrared spectral range. Besides, two well-defined small peaks at 319 and 371 nm, corresponding to individual FeCl$_3$ thin-film spectrum, and a maximum at 813 nm, are emphasized. This can be explained based on the presence of positive polaron charge carriers caused by doping (by dopant uptake) [22], or as confirming the polymer oxidation in the presence of FeCl$_3$ [16].

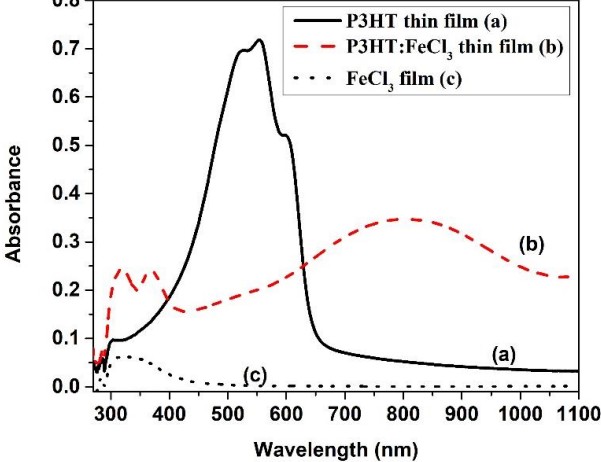

**Figure 3.** The absorption spectra.

From ellipsometric measurements, the refractive index and extinction coefficient were determined, the dispersion curves of which are given in Figure 4. The experimental ellipsometric data were simulated by using models based on New Amorphous Dispersion formula (Equations (1)–(3)), applied on the parameters from Table 1. The ellipsometric model [25] consists in a glass substrate, a sensitized polymer thin film, and an iron oxide layer.

$$n(\omega) = n_\infty + \frac{B \cdot (\omega - \omega_j) + C}{(\omega - \omega_j)^2 + \Gamma_j^2},$$ (1)

where

$$\begin{cases} B_j = \dfrac{f_j}{\Gamma_j} \cdot \left[ \Gamma_j^2 - (\omega_j - \omega_g)^2 \right] \\ C_j = 2 \cdot f_j \cdot \Gamma_j \cdot (\omega_j - \omega_g) \end{cases}$$ (2)

$$k(\omega) = \begin{cases} \dfrac{f_j \cdot (\omega - \omega_j)}{(\omega - \omega_j)^2 + \Gamma_j^2}; \ for \ \omega > \omega_g \\ 0; \qquad \quad for \ \omega < \omega_g \end{cases}$$ (3)

The above relation describes $n(\omega)$ and $k(\omega)$ as the refractive index and extinction coefficient. At the same time, $\Gamma_j$ suggests a broadening parameter of the absorption peak ($0.2 < \Gamma_j < 8$), $f_j$ represents the amplitude of the extinction coefficient ($0 < f_j < 1$), $\omega_j$ is the energy corresponding to the maximum extinction coefficient ($1.5 < \omega_j < 10$), and $\omega_g$ defines the energy corresponding to the minimum extinction coefficient ($\omega_g < \omega_j$).

**Table 1.** Ellipsometric parameters used to simulate the models.

| Sample | Dispersion Formula | Specific Parameters | | | | | Thickness (nm) | $\chi^2$ |
|---|---|---|---|---|---|---|---|---|
| | | $n_\infty$ | $\omega_g$ (eV) | $f_j$ (eV) | $\omega_j$ (eV) | $\Gamma_j$ (eV) | | |
| P3HT | 4 × NA | 0.78 ± 0.17 | 1.69 ± 0.01 | 0.01 ± 0.001 | 4.58 ± 0.1 | 0.07 ± 0.01 | 71 ± 3.6 | 3.74 |
| | | | | 0.32 ± 0.02 | 2.07 ± 0.01 | 0.25 ± 0.01 | | |
| | | | | 0.13 ± 0.05 | 6.89 ± 1.04 | 1.21 ± 0.33 | | |
| | | | | 0.11 ± 0.04 | 2.96 ± 0.09 | 0.88 ± 0.09 | | |
| P3HT: FeCl$_3$ | P3HT layer: 4 × NA | 1.81 ± 0.65 | 1.07 ± 0.01 | 0.31 ± 0.02 | 1.01 ± 0.01 | 0.15 ± 0.01 | 115 ± 1.5 | 1.58 |
| | | | | 0.52 ± 0.03 | 1.34 ± 0.02 | 0.27 ± 0.02 | | |
| | | | | 0.01 ± 0.001 | 5.96 ± 0.32 | 0.04 ± 0.008 | | |
| | | | | 0.37 ± 0.21 | 2.15 ± 0.49 | 0.34 ± 0.19 | | |
| | FeCl$_3$ layer | 1.75 ± 0.02 | 1.04 ± 0.66 | 0.001 ± 0.0001 | 1.39 ± 0.08 | 0.34 ± 0.11 | 22 ± 1.5 | |

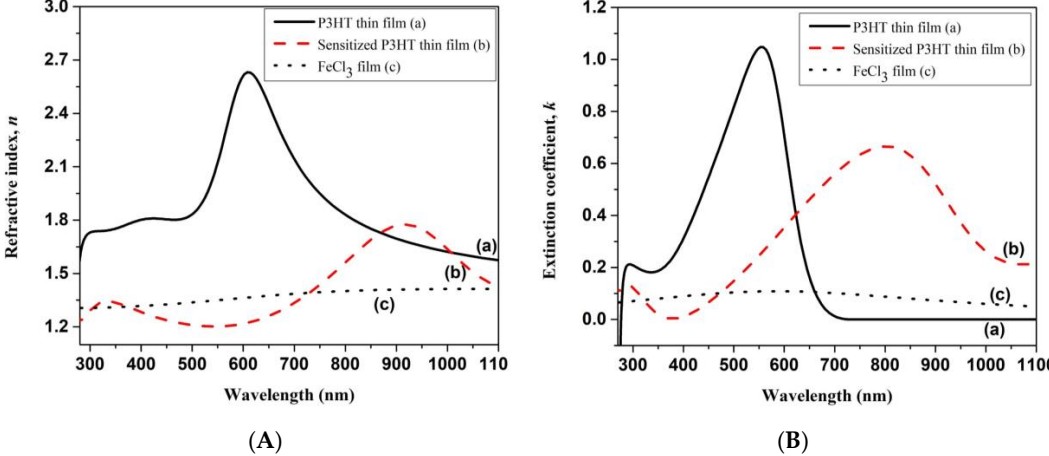

**Figure 4.** (**A**) Refractive index; (**B**) extinction coefficient.

Evaluating primarily $FeCl_3$ influence on P3HT, considering the refractive index peak of P3HT at 610 nm, $n_{max}^{P3HT} = 2.6$, after immersion (P3HT:$FeCl_3$), P3HT was found to completely disappear, besides a new absorption band with a maximum at 920 nm, $n_{max}^{P3HT:FeCl_3} = 1.7$, shows up. At the same time, P3HT:$FeCl_3$ displays a significant decrease in the refractive index values in the visible range (300 to 800 nm). Being directly proportional with the absorption coefficient, the extinction coefficient (Figure 5 right graph) exhibits the same behavior.

After $FeCl_3$ immersion of polymer thin films, their resistivity displays a marked decrease of six orders of magnitude. Figure 5 presents the time variation of resistivity of sensitized polymer thin films, with respect to pristine thin films. After six days, the samples were immersed again in $FeCl_3$ solution (second immersion), under the same conditions, an effect similar to the initial one being observed for P3HT films.

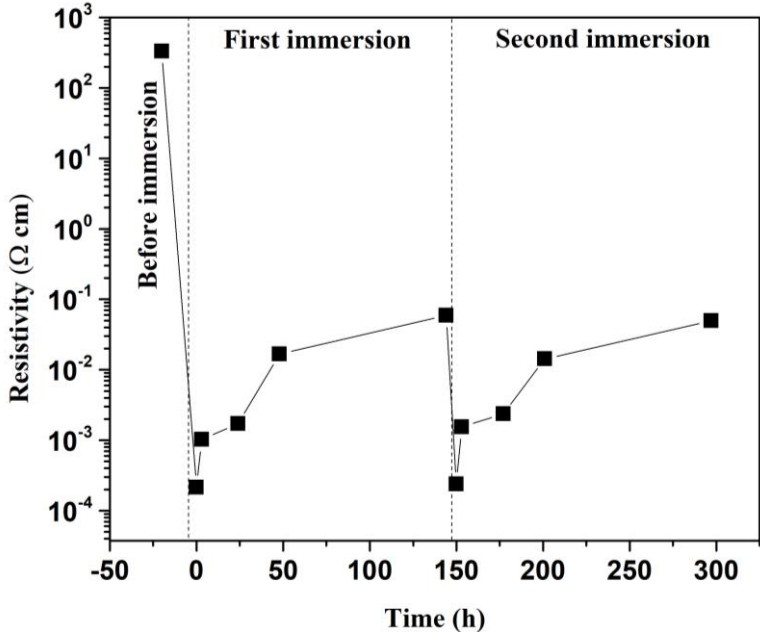

**Figure 5.** The resistivity behavior of sensitized P3HT thin films, over time.

The temperature dependence of resistivity was studied during several heating/cooling cycles (20–100 °C), under both dark ambient conditions and white-light illumination of 1000 W/m$^2$ (Figure 6). The electrical resistivity was measured in parallel to the substrate, and before being immersed in $FeCl_3$, P3HT thin films showed the same behavior as that described by Girtan [20]; besides, the light influence on resistivity is significant, consisting of a marked decrease in the resistivity value, caused by photoconduction processes (Figure 6A). Taking into account that charge carrier transport occurs via hopping mechanisms between $\pi$–$\pi$ polymeric chain domains, the heating and cooling cycles are not reversible, suggesting that the polymer requests a resting time to adjust or some inertial mechanisms take place. After immersion (Figure 6B–D), the temperature dependence of resistivity exhibits a similar allure; the presence of light does not lead to a decrease in resistivity, as in the pristine case, but on the contrary, the resistivity increases upon light exposure. After the second $FeCl_3$ immersion (Figure 6D), the polymer thin film displays the same behavior and similar resistivity values to the previous case, after the first immersion (Figure 6B), conferring the reproducibility and repeatability of the measurements.

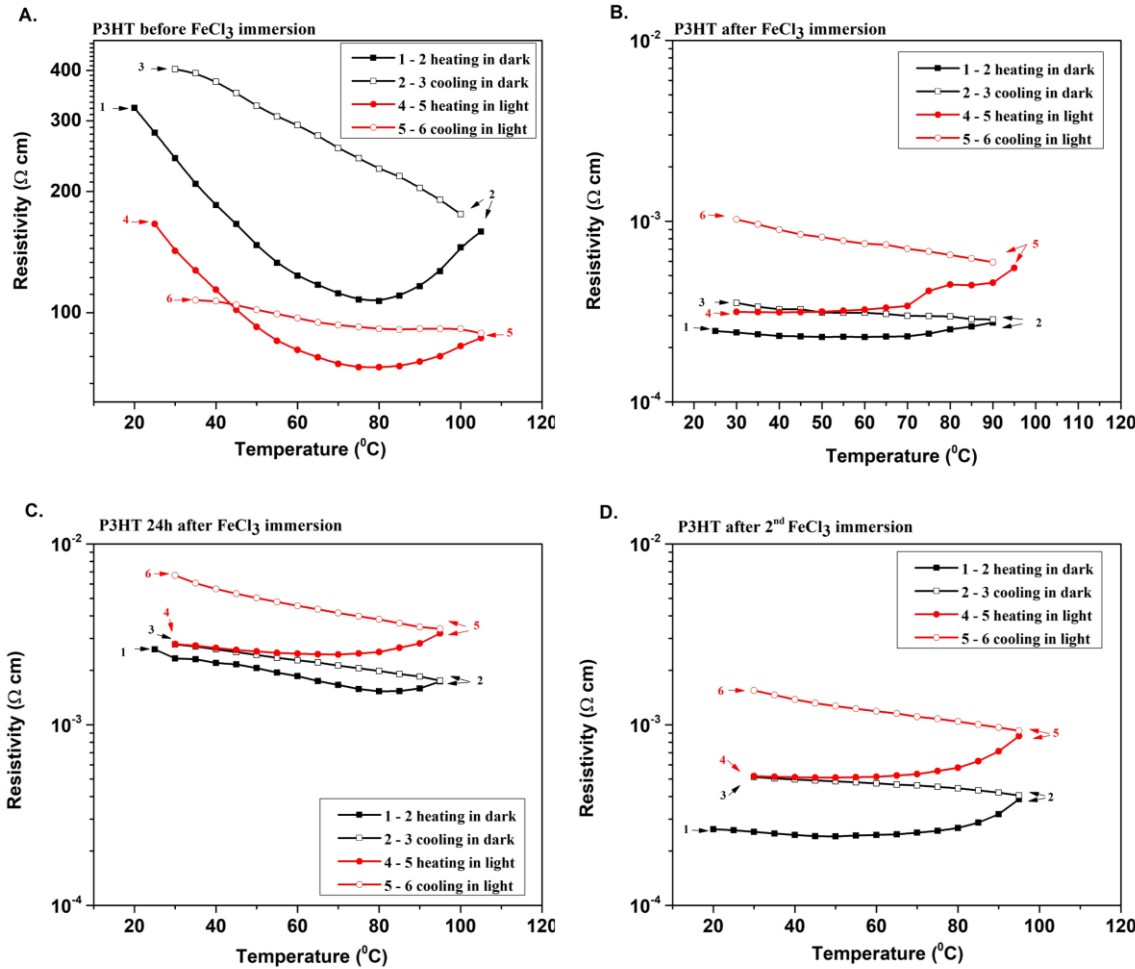

**Figure 6.** Temperature dependence of resistivity during heating (filled dots) and cooling (empty dots) cycles, under dark (black squares) and light (red stars) conditions: (**A**) before $FeCl_3$ immersion, (**B**) immediately after immersion, (**C**) 24 h after immersion, and (**D**) after the second immersion).

## 4. Conclusions

The effect of $FeCl_3$ sensitization on physical properties of P3HT thin films is presented. The study of electrical properties emphasizes the behavior of $FeCl_3$ immersed polymer films, depending on time and temperature. A high decrease in the electrical resistivity was shown immediately after the immersion, from $10^3$ to $10^{-4}$ $\Omega \cdot cm$, but the resistivity tends to return slowly to the original value: In six days, the resistivity passes from $10^{-4}$ to $10^{-2}$ $\Omega \cdot cm$. When we repeat the immersion process on the same sample, the effect remains the same. Concerning the thermal stability and depending on procedure (time of immersion), the resistivity values are between $10^{-4}$ and $10^{-3}$ $\Omega \cdot cm$, with the exception of the as-prepared polymer sample, which measures $10^3$ $\Omega \cdot cm$. The reproducibility of the phenomenon was revealed performing several immersion processes of the same sample, and, additionally, the time-dependence behavior was remarked, along with the thermal stability of the sensitized films.

Using $FeCl_3$ sensitization, P3HT thin films with enhanced electrical properties, suitable for solar cells applications, can be obtained.

**Author Contributions:** Conceptualization, M.G. and L.L.; methodology, L.H., M.D., and C.D.; software, L.H.; validation, M.G. and L.L.; formal analysis, L.H., M.G., and L.L.; investigation, L.H., M.D., and C.D.; resources, M.G., L.L., M.D., and C.D.; data curation, L.H.; writing—original draft preparation, L.H.; writing—review and editing, L.L. and M.G.; visualization, M.G.; supervision, M.G. and L.L.; project administration, M.G.; funding acquisition, L.H. All authors have read and agreed to the published version of the manuscript.

**Funding:** This work was funded by the European Social Fund, through Operational Programme Human Capital 2014–2020, project number POCU/380/6/13/123623, project title "PhD Students and Postdoctoral Researchers Prepared for the Labour Market", by ERASMUS + Program and by Bourse du Gouvernement Francais (Campus France).

**Conflicts of Interest:** The authors declare no conflict of interest.

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
