# Peer review of "On the Electrical and Optical Properties Stability of P3HT Thin Films Sensitized with Nitromethane Ferric Chloride Solutions"

_coatings, doi:10.3390/coatings10111074_

Round 1

Reviewer 1 Report

Dear Authors,

I have read the manuscript entitled  On the electrical and optical properties stability of P3HT thin films  sensitized with nitromethane ferric chloride solutions and I found results presented in this work to be interesting, nevertheless that P3HT doped films are well-known and widely presented in the literature.    

Now, I would like to let you know that my opinion is rather positive and I consider that this manuscript can be published in the Special Edition of Coatings . However, some doubts and comments should be taken into consideration and a few minor changes have to be done:

  • -Introduction

The full names of: SCUT-CSU, PBDB-T, PBDB-T-SF, F8T2 and PCDTBT should be introduced;

  • Materials and Methods

What was the thickness of P3HT film after doping and after repeating the immersion process?

Why the thickness was obtained by profilometry, not using elipsometric measurements ?

  • -Results

UV-Vis absorption spectra (Figure 3) of these type of films are presented and discussed in many papers, for example in [16,17]. According to [17], the spectrum of P3HT:FeCl3 can be explained by positive polaron, not by bipolaron form, as suggest authors !!  

  • Conclusions

please extend the summary in the context of photovoltaic applications. What is time and thermal stability of these films, used in solar cells.

As final conclusion,  I recommend this manuscript to be published in Coatings, after these corrections, mentioned above.     

Author Response

Referee 1:

  • -Introduction

A: lines 35 – 40: In agreement with reviewer’s suggestions, the full names of: SCUT-CSU, PBDB-T, PBDB-T-SF, F8T2 and PCDTBT were introduced;South China University of Technology (SCUT);Poly[(2,6-(4,8-bis(5-(2-ethylhexyl)thiophen-2-yl)-benzo[1,2-b:4,5-b’]dithiophene))-alt-(5,5-(1’,3’-di-2-thienyl-5’,7’-bis(2-ethylhexyl)benzo[1’,2’-c:4’,5’-c’]dithiophene-4,8-dione)] (PBDB-T) or its derivatives, Poly(9,9-dioctylfluorene-alt-bithiophene) (F8T2) or Poly[N-9'-heptadecanyl-2,7-carbazole-alt-5,5-(4',7'-di-2-thienyl-2',1',3'-benzothiadiazole)] (PCDTBT).

  • Materials and Methods

What was the thickness of P3HT film after doping and after repeating the immersion process?

A: The thickness of the layers was increased by approx. 20 nm.

Why the thickness was obtained by profilometry, not using elipsometric measurements ?

A: The thickness was also determined by ellipsometric measurents as it can be seen now in Table 1 (line 120).

  • Results

UV-Vis absorption spectra (Figure 3) of these type of films are presented and discussed in many papers, for example in [16,17]. According to [17], the spectrum of P3HT:FeCl3 can be explained by positive polaron, not by bipolaron form, as suggest authors !!  

A: The authors thank to reviewer for this observation, the necessary correction was added on line 99.

  • Conclusions

please extend the summary in the context of photovoltaic applications. What is time and thermal stability of these films, used in solar cells.

A: According to reviewer’s suggestion, more precisions were added on lines 162 - 164: concerning the thermal stability and depending on procedure (immersion moment), the resistivity values range between 10-4 and 10-3 Ω×cm, except for the as-prepared polymer sample, whose resistivity is 103 Ω×cm.

Reviewer 2 Report

The manuscript of Hrostea et al. reports on the of optical and electrical properties stability of P3HT:FeCl3 thin films. This study is interesting and relatively comprehensive. Thus, I think it will attract some attention from researchers in related areas. However, there are several major and minor improvements needed before accepting this manuscript for publication. Detailed comments and questions are listed as follows:

Minor concerns:

The styling of the figures has to be improved:

  1. Figure 1 does not show “Electrical measurement configuration.” It only shows a sample configuration. Ether figure caption has to be changed, or the figure should show more details about the measurement setup.
  2. Figures 4 and 5 (refractive index and extinction coefficient) can be easily merged in one figure (two panels in one row) as they show more or less complementary data. Moreover, the descriptions of these figures are combined in the text, which makes it even more convincing to have them as single figure.
  3. The legends in each figure should stay inside the figure frame
  4. The data presented in Figure 7 will be much easier to compare if panels would be arranged as 2x2 matrix and the scales of the y-axis (resistivity)´would be the same for all 4 panels (or at least for panels B-D)
  5. Figure 6 shows a change of resistivity over time. There 2 cycles are presented, but the way it is presented is misleading. From the figure, it is not very clear if it was done on one sample, or it is 2 different samples, which shows similar results. As it was continuous experiments for 12 days, I would suggest to use x scale 0-12 days and mark with arrows the day at which sensitization was done. Also, place a mark 1st point, which shows resistivity before sensitization, and clearly state about this first point in the text.

Materials and Methods have to be improved:

  1. The description of sensitization with FeCl3 is missing. Although authors have used protocols from the literature, it is absolutely not convenient to look for protocol in different paper. Moreover, no details about sensitization are given. What was the concentration of FeCl3 in nitromethane? Any washing, drying procedures? Please provide details as it is a significant part of the paper.
  2. Details of how the n and k were determined are also missing. Which model was used and fitting parameters?

Major concerns:

  1. First of all, the whole paper only describes the presented figures, and discussion of obtained results is absolutely missing. There is not even an attempt to describe the physics behind observed results. Why are optical constants changing in this particular way? Any comparison for existing cases in literature? The same holds for resistivity measurements. A small discussion and explanation of results are necessary.
  2. I understand that there is not much new physics, and it is more a systematic study, which is also very important. Nevertheless, any systematic study must contain statistics. How many samples were measured? For example, in line 55 the thickness of polymer film is stated “96 nm” such statement is appropriate when only one sample was measured. If there were more than one, mean+- SD should be given. The same holds for resistivity measurements (Figure 6). The graph shows only 11 points measured for 1 sample. How reproducible this measurement? How many samples were measured? A plot with mean and SD is necessary here as well.
  3. The authors aim “to investigate the reproducibility and the stability over time and during many cycles of heating and cooling, in dark and under illumination, of the electrical properties of sensitized polymer films”, however, there are no data presenting “many” cycles of heating and cooling. Figure 7 shows only one cycle (heating+cooling) for each experimental condition. Was there more than one cycle measured? If yes, please show it. If not, then sentences in the introduction must be changed accordingly.
  4. Figure 2 and figure 5 are repeating each other, which is also mentioned by authors in the text! I found the figure 2 to be inessential in the manuscript. It does not provide any additional information compared to figure 5.
  5. The extinction coefficient presented in figure 5 is negative for a wavelength above 700 nm. This is physically incorrect. It is an obvious error either in measurement or data processing. It is absolutely unacceptable to present it in such a way in publication.

Author Response

Referee 2:

  1. Figure 1 does not show “Electrical measurement configuration.” It only shows a sample configuration. Ether figure caption has to be changed, or the figure should show more details about the measurement setup.

A: In agreement with reviewer’s suggestion, the figure caption was changed (line 79): Sample configuration (used for electrical measurements).

  1. Figures 4 and 5 (refractive index and extinction coefficient) can be easily merged in one figure (two panels in one row) as they show more or less complementary data. Moreover, the descriptions of these figures are combined in the text, which makes it even more convincing to have them as single figure.

A: line 121: This change was done.

  1. The legends in each figure should stay inside the figure frame

A: line 101, 121, 136, 152: This change was done.

  1. The data presented in Figure 7 will be much easier to compare if panels would be arranged as 2x2 matrix and the scales of the y-axis (resistivity)´would be the same for all 4 panels (or at least for panels B-D)

A: line 152: This change was done.

  1. Figure 6 shows a change of resistivity over time. There 2 cycles are presented, but the way it is presented is misleading. From the figure, it is not very clear if it was done on one sample, or it is 2 different samples, which shows similar results. As it was continuous experiments for 12 days, I would suggest to use x scale 0-12 days and mark with arrows the day at which sensitization was done. Also, place a mark 1stpoint, which shows resistivity before sensitization, and clearly state about this first point in the text.

A: line 136: This change was done.

Materials and Methods have to be improved:

  1. The description of sensitization with FeCl3 is missing. Although authors have used protocols from the literature, it is absolutely not convenient to look for protocol in different paper. Moreover, no details about sensitization are given. What was the concentration of FeCl3 in nitromethane? Any washing, drying procedures? Please provide details as it is a significant part of the paper.

A: line 61 – 64: The sensitization with FeCl3 was carried out as described in [22] by samples submersion in a 5 g/L nitromethane solution during one hour. The submersed samples were dried freely in the normal atmosphere and no chemical agent surplus cleaning procedures were applied.

  1. Details of how the n and k were determined are also missing. Which model was used and fitting parameters?

A: The ellipsometric measurements procedure is shown from line 104 to 130.

Major concerns:

  1. First of all, the whole paper only describes the presented figures, and discussion of obtained results is absolutely missing. There is not even an attempt to describe the physics behind observed results. Why are optical constants changing in this particular way? Any comparison for existing cases in literature? The same holds for resistivity measurements. A small discussion and explanation of results are necessary.

A: The discussion about resistivity was developed at lines 140 – 146.

  1. I understand that there is not much new physics, and it is more a systematic study, which is also very important. Nevertheless, any systematic study must contain statistics. How many samples were measured? For example, in line 55 the thickness of polymer film is stated “96 nm” such statement is appropriate when only one sample was measured. If there were more than one, mean+- SD should be given. The same holds for resistivity measurements (Figure 6). The graph shows only 11 points measured for 1 sample. How reproducible this measurement? How many samples were measured? A plot with mean and SD is necessary here as well.

A: Regarding the thickness of the sample, based on the reviewer’s suggestion, on line 59, this value is stated now with the standard deviation. This study was done on several samples performing different sets of electrical measurements. The reproducibility is highlighted in time (the evolution of the sample after chemical sensitization), but also repeating the measurements several times for the same moment of evolution. In addition, because of the movement of polymeric chains inside of the layer, any mean value of resistivity was calculated. 

  1. The authors aim “to investigate the reproducibility and the stability over time and during manycycles of heating and cooling, in dark and under illumination, of the electrical properties of sensitized polymer films”, however, there are no data presenting “many” cycles of heating and cooling. Figure 7 shows only one cycle (heating+cooling) for each experimental condition. Was there more than one cycle measured? If yes, please show it. If not, then sentences in the introduction must be changed accordingly.

A: line 53: during heating/cooling cycles

  1. Figure 2 and figure 5 are repeating each other, which is also mentioned by authors in the text! I found the figure 2 to be inessential in the manuscript. It does not provide any additional information compared to figure 5.

A: Figure 2 – Absorbance can be considered as a validation of the new figure 4, representing the extinction coefficient, which was done after the ellipsometric model simulation. Figure 2 confirms that the model is valid.

  1. The extinction coefficient presented in figure 5 is negative for a wavelength above 700 nm. This is physically incorrect. It is an obvious error either in measurement or data processing. It is absolutely unacceptable to present it in such a way in publication.

A: Taking in account reviewer’s suggestion, the ellipsometric models were reexamined and the error was corrected as it can be seen on line 121.

Round 2

Reviewer 2 Report

The last sentence in Conclusion section state:

"However, this effect is not persistent in the case of thinner films."

Since there were no comparison between samples with different thickness, authors should give appropriate literature reference to support this claim. Or delete this sentence. 

Author Response

Thank you very much for your suggestion. The last sentence was removed from the manuscript.